# Pulmonary Function in Post-Infectious Bronchiolitis Obliterans in Children: A Systematic Review and Meta-Analysis

**DOI:** 10.3390/pathogens11121538

**Published:** 2022-12-14

**Authors:** Eun Lee, Suyeon Park, Hyeon-Jong Yang

**Affiliations:** 1Department of Pediatrics, Chonnam National University Hospital, Chonnam National University Medical School, Gwangju 61469, Republic of Korea; 2Department of Applied Statistics, Chung-Ang University, Seoul 06974, Republic of Korea; 3Department of Biostatistics, Soonchunhyang University College of Medicine, Seoul 04401, Republic of Korea; 4Department of Pediatrics, Soonchunhyang University Seoul Hospital, Soonchunhyang University College of Medicine, Seoul 04401, Republic of Korea

**Keywords:** children, post-infectious bronchiolitis obliterans, pulmonary function tests

## Abstract

Owing to the rarity of post-infectious bronchiolitis obliterans (PIBO), pulmonary function in children with PIBO has been mainly investigated in small-sample sized studies. This systematic review and meta-analysis investigated pulmonary function in children with PIBO, regardless of age at respiratory insult and PIBO diagnosis. A systematic literature search revealed 16 studies reporting pulmonary function data in 480 children with PIBO. Levels of key pulmonary function parameters were summarized by pooled mean difference (MD) only in children with PIBO, and a random effect model was used. Pooled MDs (95% confidence interval [CI]) for FEV_1_, FVC, and FEF_25–75%_ were 51.4, (44.2 to 58.5), 68.4 (64.4 to 72.3), and 26.5 (19.3 to 33.6) % predicted, respectively, with FEV_1_/FVC of 68.8% (62.2 to 75.4). Pooled MDs (95% CI) of the z-scores for FEV_1_, FVC, and FEF_25–75%_ were −2.6 (−4.2 to −0.9), −1.9 (−3.2 to −0.5), and −2.0 (−3.6 to −0.4). Pooled MD (95% CI) for the diffusion capacity of the lungs for carbon monoxide from two studies was 64.9 (45.6 to 84.3) % predicted. The post-bronchodilator use change in the FEV_1_ in three studies was 6.1 (4.9 to 7.2). There was considerable heterogeneity across the studies. PIBO is associated with moderately impaired pulmonary function, and this review facilitates an understanding of PIBO pathophysiology in children.

## 1. Introduction

Post-infectious bronchiolitis obliterans (PIBO) is one of the various rare complications of a lower respiratory tract infection [1]. Although PIBO can develop at all ages in patients ranging from infants to adults, PIBO is much more common in children [2]. Various respiratory pathogens, including adenovirus and *Mycoplasma pneumoniae*, are associated with PIBO development, and the causative pathogens are unknown in some cases [2,3,4]. Due to its chronic course of the disease, PIBO confers a substantial disease burden [2,5].

Although the exact immunopathologic mechanisms of PIBO have not been fully elucidated, PIBO is characterized by peri-bronchiolar fibrosis in the small airways, which results in various degrees of irreversible airway obstruction [6]. In addition, patients with PIBO show varying degrees of severity with diverse clinical outcomes. Pulmonary function tests can be helpful in the diagnosis of PIBO, the monitoring of disease status, and the prediction of prognosis in children with PIBO [7]. In addition, regular follow-up of pulmonary function can provide information on pulmonary growth with regard to small airway pathology in children with PIBO [2]. However, pulmonary function tests using spirometry are generally impossible to perform at the time of development or diagnosis of PIBO due to the lack of corporation of young patients. Therefore, children with PIBO perform pulmonary function tests when they attain an age at which pulmonary function tests are possible.

The previous studies on the various pulmonary function parameters in children with PIBO showed a diverse range of impairment due to differences in the characteristics of the study population and the degree and age of respiratory insults [8,9,10,11,12,13,14]. However, all studies used small sample sizes due to the rarity of the disease [2]. Additionally, no systematic review and meta-analysis on pulmonary function in children with PIBO has been conducted.

We hypothesized that PIBO in early life can affect pulmonary function even in childhood. The aim of the present study was to investigate the levels of key pulmonary function parameters in children with PIBO, regardless of the age at development and the diagnosis of PIBO and the degree of respiratory insult.

## 2. Methods

### 2.1. Literature Search Strategy

This systematic review and meta-analysis were performed according to the Preferred Reporting Items for Systematic Review and Meta-Analyses (PRISMA) reporting guidelines [15]. We searched the PubMed, Cochrane Library, and Embase databases for relevant studies from inception until 2 June 2022 by using a combination of search terms of “bronchiolitis obliterans” and “pulmonary function”. This meta-analysis study was not registered.

### 2.2. Eligibility and Exclusion Criteria

The inclusion criteria for the study selection were the following: (1) pediatric patients with PIBO; (2) original articles reporting randomized controlled trials, studies with retrospective and prospective designs, case-control studies, and case series, which included information on the diverse pulmonary function parameters, regardless of age at the development and diagnosis of PIBO, severity of PIBO, and intervals between the index date of PIBO development and the date when pulmonary function was performed; and (3) PIBO diagnosed based on clinical features, chest radiography, and chest computed tomography after the exclusion of other causes of chronic pulmonary disease [2]. The exclusion criteria were the following: (1) inclusion of any adult patients (age > 18 years) with PIBO as well as both adults and children with PIBO, to minimize the heterogeneity of the study population; (2) studies that included patients with other respiratory comorbidities and other underlying diseases except PIBO, such as cystic fibrosis; (3) any case report that included only one case of PIBO; (4) abstracts; and (5) studies that did not include key pulmonary function parameters, such as forced expiratory volume in the first second (FEV_1_), as well as forced vital capacity (FVC).

### 2.3. Study Selection and Data Extraction

The study selection was performed independently by two reviewers (EL and HJY), and disagreements, if any, were resolved by consensus. Study data were extracted using a standardized data extraction form that included the author’s name; year of publication; country of study; definition of PIBO; study population size; patients’ demographics (sex and age at performance of pulmonary function test); and values of pulmonary function parameters, including FEV_1_, FVC, FEV_1_/FVC, forced expiratory flow between 25% and 75% of vital capacity (FEF_25–75%_), and diffusing capacity for carbon monoxide (DL_CO_). In cases where the study populations were divided into two subgroups, a pooled mean and standard deviation (SD) for the combined subgroup populations was calculated [8,16]. The presented pulmonary function parameters differed according to each study and were expressed as diverse values, such as the % predicted, z-score, or liters.

### 2.4. Study Quality

The risk of bias for the studies included in the present meta-analysis was assessed using the National Institutes of Health Quality Assessment Tool for Observational Cohort and Cross-Sectional Studies [17]. This tool includes 14 items on details of the population, participation rate, sample size, and potential confounding factors. The two authors (EL, EMY and HJY) evaluated the quality of each study by assigning an overall rating of good, fair, or poor.

### 2.5. Data Synthesis and Statistical Analysis

The assessment of publication bias was performed using the Egger test, and analyses for the presence of potential outliers were performed. We used random-effects meta-analyses to calculate the mean difference (MD) with a 95% confidence interval (CI). The statistical analysis was performed using Review Manager version 5.3 (Cochrane Collaboration, Baltimore, MD, USA). To assess the heterogeneity and inconsistency between the studies, *I*^2^ statistics were applied. *I*^2^ > 50% was defined as significant heterogeneity. 

## 3. Results

### 3.1. Study Selection

After eliminating duplicate and irrelevant articles, 30 articles were identified. Of these, we excluded 17 articles owing to the following reasons: seven were abstract-only publications, five could not be found, two provided insufficient numerical data for inclusion in the meta-analysis, one had an overlapping population, and two did not include the key pulmonary function parameters. Finally, we included 16 studies in this meta-analysis. Figure 1 shows the PRISMA flow chart. All studies presented pulmonary function parameters in children with PIBO. In all of the included studies, PIBO was diagnosed based on the clinical features and chest computed tomography findings with pulmonary function tests, if possible, after ruling out other causes of chronic pulmonary diseases [2].

### 3.2. Study Characteristics

The characteristics of the included studies are described in Table 1. Five studies were conducted in Brazil [13,16,18,19,20], two in Korea [8,11], two in Argentina [9,21], two in China [22,23], one in USA [14], one in Italy [24], one in Israel [10], one each in Brazil and Chile [12], and another one each in USA and Korea [25]. Seven studies were cross sectional studies [8,9,11,12,13,23,25], whereas another six studies were case series [14,18,20,21,22,24]; two studies were case-control studies [10,19], and one study was a randomized controlled trial [16]. A total of 480 children with PIBO were included in this study, and a pulmonary function test was performed at the age of 4–17 years.

### 3.3. Pulmonary Function in PIBO

Here, we presented data on key pulmonary function parameters and lung volume in children with PIBO, and the details are provided in Appendix A.

### 3.4. FEV_1_ and FVC

A total of ten studies [10,12,13,16,19,20,21,23,24,25] reported data on the FEV_1_ and FVC % predicted, and five studies [8,9,10,11,18] included data on the z-score of FEV_1_ and FVC. The pooled estimates for FEV_1_ % predicted were 51.4% (95% CI, 44.2% to 58.5%) when more than 80% was considered as a reference (Figure 2a). The pooled estimates for the z-score of FEV_1_ were −2.6 (95% CI, −4.2 to −0.9) (Figure 2b). The pooled estimates for the FVC % predicted were 68.4% (95% CI, 64.4% to 72.3%) when more than 80% was considered as a reference (Figure 3a). The pooled estimates for the z-score of FVC was −1.9 (95% CI, −3.2 to −0.5) (Figure 3b). Only two studies [13,19] measured the FEV_1_ and FVC in liters, and the pooled estimates for the FEV_1_ and FVC were 1.4 L (95% CI, 0.4 to 2.4) and 2.2 L (95% CI, 1.2 to 3.3), respectively (Appendix A).

### 3.5. FEV_1_/FVC

A total of eight studies [12,13,16,19,20,21,23,24] reported data on the FEV_1_/FVC %, and four studies [8,9,11,18] included data on the z-score of the FEV_1_/FVC. The pooled estimates for the FEV_1_/FVC % were 68.8% (95% CI, 62.2% to 75.4%) when more than 70% of FEV_1_/FVC (%) was considered as a reference (Figure 4a). The pooled estimates for the z-score of FEV_1_/FVC was −2.0 (95% CI, −2.5 to −1.5) (Figure 4b).

### 3.6. FEF_25–75%_

Eight articles [10,12,13,16,20,21,24,25] reported the FEF_25–75%_ % predicted, whereas four articles [8,9,10,11] presented the z-score of FEF_25–75%_. The pooled estimates for the FEF_25–75%_ % predicted were 26.5% (95% CI, 19.3% to 33.6%) (Figure 5a). The pooled estimates for the z-score of FEF_25–75%_ were −2.0 (95% CI, −3.6 to −0.4) (Figure 5b).

### 3.7. DL_CO_

Two studies [19,24] reported the DLco % predicted (Appendix A). The pooled estimate of the DLco % predicted was 64.9% (95% CI, 45.6% to 84.3%) in children with PIBO.

### 3.8. Bronchodilator Response

A total of three studies [8,11,20] mentioned the prevalence of a bronchodilator response, defined as an increase in the FEV_1_ by at least 12% following salbutamol inhalation. The prevalence of positive bronchodilator responses in children with PBIO ranged from 30% to 83.3% [20,22,24]. A total of three studies [8,11,20] reported data on the degree of FEV_1_ change after salbutamol inhalation, while levels of the % change in the FEV_1_ in the two groups were merged in one study [8]. The pooled estimate for the bronchodilator response in the FEV_1_ % predicted from the two studies was 6.1% (95% CI, 4.9% to 7.2%).

### 3.9. Parameters of Diverse Lung Volume

We presented the pooled estimates of the diverse lung volume parameters in the Appendix A.

### 3.10. Publication Bias

There was no significant publication bias in the included studies.

## 4. Discussion

The present systematic review and meta-analysis showed levels of key pulmonary function parameters in children with PIBO, regardless of age at development and diagnosis of PIBO. The degree of decrease in the levels of pulmonary function parameters was greater in the FEF_25–75%_ and FEV_1_, followed by the FEV_1_/FVC and FVC compared to the normal standards in each parameter for age. A positive bronchodilator response was seen in 30–83.3% of children with PIBO. As this is the first systematic review and meta-analysis on pulmonary function in children with PIBO, the present study is meaningful for providing information on the key parameters of pulmonary function in children with PIBO and therefore enables the prediction of prognosis with regard to the impairment of pulmonary function during childhood.

The time interval between respiratory insults and PIBO development is diverse [20]. It is often impossible to estimate the exact time of respiratory insults that result in PIBO due to the frequent respiratory infections in early life and, in some cases, PIBO development after non-severe respiratory infections. The duration of persistent airway inflammation in patients with PIBO is neither clearly identified, nor is the effect of airway inflammation in PIBO on long-term pulmonary function clearly known, partially due to heterogeneities in severity, onset age of PIBO, and diverse host factors [14,20,24]. Therefore, the investigation of the pulmonary function in children, regardless of the heterogeneities in the characteristics of the patients with PIBO, is helpful in terms of monitoring and predicting the prognosis of PIBO in children.

The differences in the degree of impairment in each pulmonary function parameter compared to normal reference levels reflect the pathophysiology of PIBO in children with long-term clinical courses. In the present meta-analysis, when considering the normal standards for age (≥65% of predicted), the FEF_25–75%_ levels were maximally decreased in children with PIBO [26,27]. The substantial decrease in the FEF_25–75%_ levels, which is more sensitive than the FEV_1_ level in detecting small airway obstructions [28], suggests the prominent dysfunction of the small airways in children with PIBO. These findings correspond to pathologic features of typical PIBO, which most commonly involves the bronchiole but not the alveoli ducts and alveoli [6]. Additionally, the range of FEF_25–75%_ levels was wider than that of other key pulmonary parameters, suggesting a diverse spectrum of small airway dysfunctions. Considering the pathophysiology of PIBO, follow-ups of pulmonary function, especially FEF_25–75%_ levels, are required for the monitoring of disease progression.

The levels of the FEV_1_ % predicted were decreased followed by the FVC % predicted. These findings suggest that the effects of respiratory insults in PIBO are more prominent in the airway than in the lung parenchyma in children. In addition, the respiratory insults in early life have a lasting impact during long-term follow-up. The present study included the cross-sectional aspect of pulmonary function without considering its long-term trajectories in children with PIBO. Therefore, the results cannot present the effects of airway and lung parenchyma growth trajectories in children with PIBO after respiratory insults. Nevertheless, the identification of the effect of respiratory insults in early life on the pulmonary function in children with PIBO enables the monitoring of respiratory health and the prediction of long-term prognosis in children with PIBO.

In the present systematic review and meta-analysis, we did not include a comparison of diverse pulmonary function parameters between children with PIBO and control groups or children with other respiratory diseases because most of the included studies reported levels of pulmonary function in children with PIBO, with the exception of one study, which compared pulmonary function in patients with PIBO to those in patients with cystic fibrosis [10]. Lung growth in early life and the potential effect of respiratory insults due to respiratory infection on lung development might differently affect the levels of diverse pulmonary function parameters between children and adults. Therefore, we included reports on pulmonary function in children with PIBO while excluding adult studies. Nevertheless, information on the levels of the key pulmonary function parameters identified in the present meta-analysis can facilitate the monitoring and prediction of the long-term prognosis in children with PIBO.

There are no standard treatment strategies in PIBO, and close monitoring for pulmonary function deterioration and the early treatment of acute infection are recommended [14]. As PIBO is characterized by fixed and irreversible airflow limitation [6], inhaled bronchodilators and/or inhaled corticosteroids have not been recommended. The irreversible airway obstruction in PIBO is characterized by an absent bronchodilator response [1]. However, according to the results of the present study, some of children with PIBO have positive bronchodilator responses with a diverse degree in each patient. This finding supports the use of bronchodilators in patients with PIBO [29]. However, because not all patients have positive bronchodilator responses among children with PIBO, the use of bronchodilators in patients with PIBO needs to be tailored to each patient. The effectiveness of inhaled corticosteroids in children with PIBO might be partially associated with the positive bronchodilator responses and airway inflammation associated with PIBO. Furthermore, the use of inhaled corticosteroids during long-term follow-up or after the acute stage of PIBO requires a tailored approach based on the characteristics of each PIBO patient.

This study has several limitations that need be noted. There is a wide range of disease heterogeneity and severity in PIBO when considering the differences in causative respiratory pathogens, age at PIBO development, and degree of respiratory insults. Although the heterogeneity of PIBO can affect the levels of diverse pulmonary function parameters, we did not consider the heterogeneity of PIBO across the studies on pulmonary function. In addition, because the age at the time of pulmonary function testing can affect the results, we did not consider the onset age of PIBO in the interpretation of the results. However, we included studies on pulmonary function only in children with PIBO and excluded studies in adults to reduce the potential confounding effect. The present study included the results of pulmonary function at specific time points; therefore, the present study has limitations in providing information on the chronologic pulmonary function changes in children with PIBO.

In conclusion, PIBO is associated with moderately impaired pulmonary function, especially small airway dysfunction. The present systematic review and meta-analysis provides information on the levels of key pulmonary function parameters in children with PIBO, regardless of age at diagnosis, the development of PIBO, or the severity of respiratory insults. Longitudinal studies of pulmonary function are required to investigated the outcomes of PIBO in children.

## Figures and Tables

**Figure 1 pathogens-11-01538-f001:**
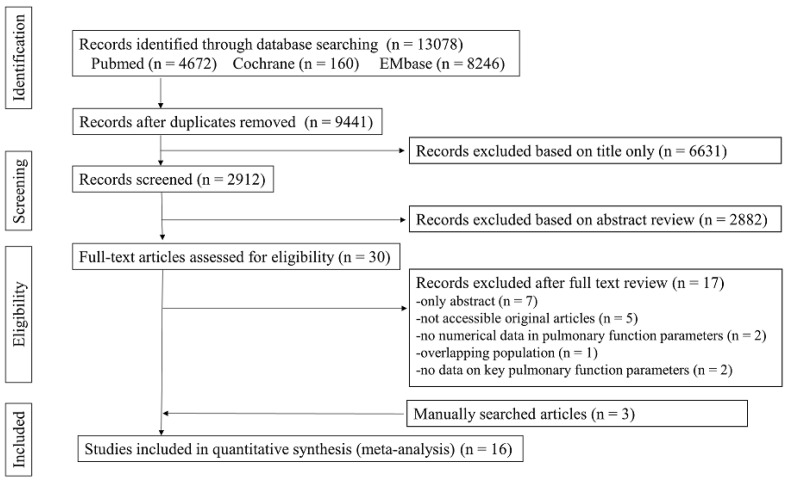
Preferred Reporting Items for Systematic Review and Meta-Analyses (PRISMA) Diagram for the Literature Search and Study Selection.

**Figure 2 pathogens-11-01538-f002:**
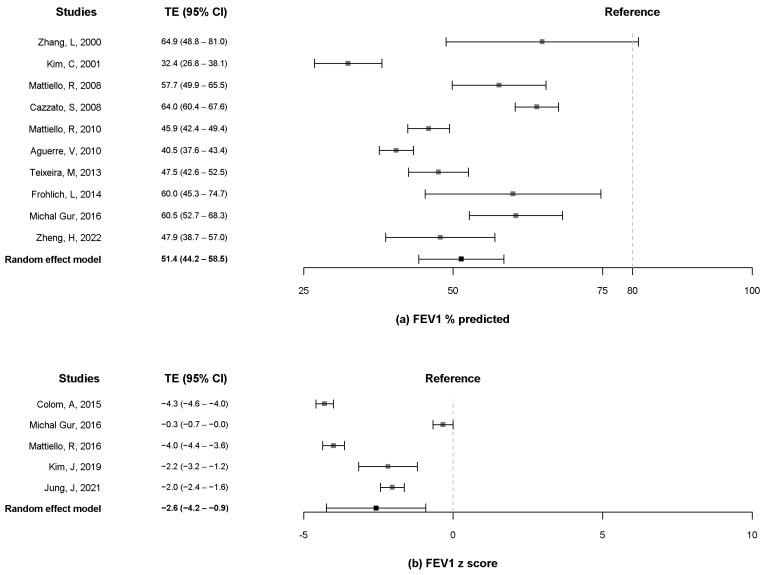
Forest plot of FEV_1_ % predicted (**a**) and z-score of FEV1. (**b**) FEV1, forced expiratory volume in 1 s [8,9,10,11,12,13,16,18,19,20,21,23,24,25].

**Figure 3 pathogens-11-01538-f003:**
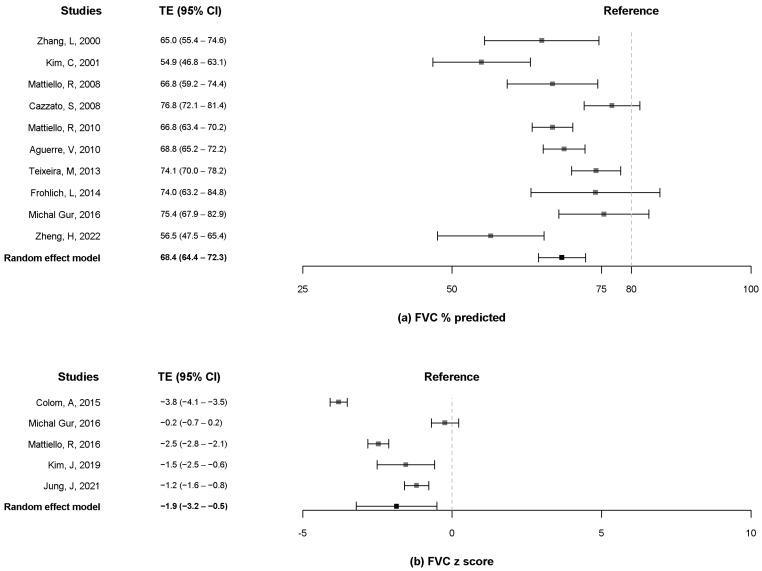
Forest plot of FVC % predicted (**a**) and z-score of FVC. (**b**) FVC, forced vital capacity [8,9,10,11,12,13,16,18,19,20,21,23,24,25].

**Figure 4 pathogens-11-01538-f004:**
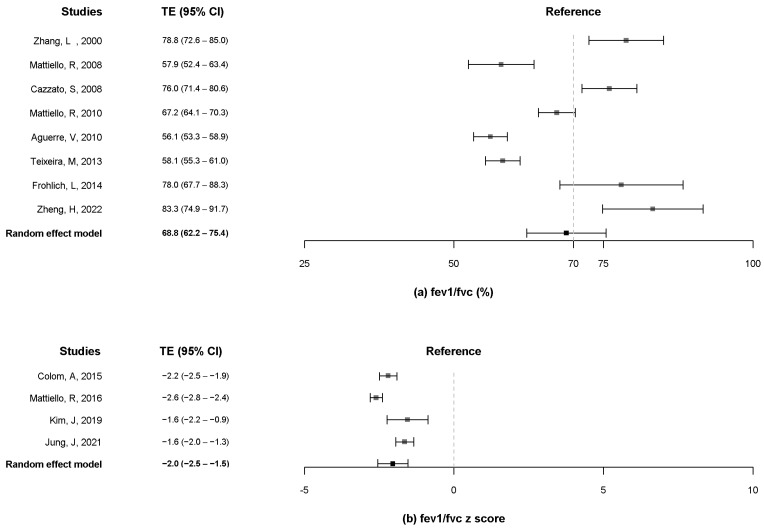
Forest plot of FEV_1_/FVC % (**a**) and z-score of FEV_1_/FVC. (**b**) FEV_1_, forced expiratory volume in one second; FVC, forced vital capacity [8,9,10,11,13,16,18,19,20,21,23,24].

**Figure 5 pathogens-11-01538-f005:**
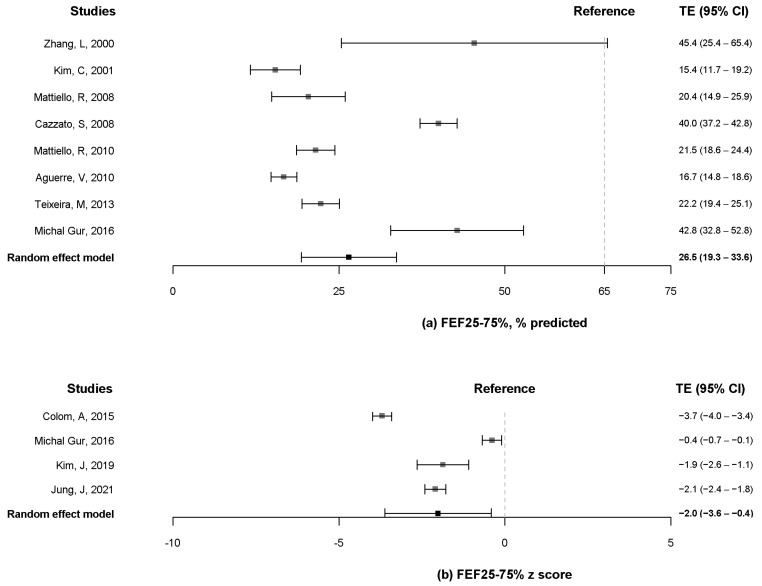
Forest plot of FEF_25–75%_ % predicted (**a**) and z-score of FEF_25–75%_. (**b**) FEF_25–75%_, forced expiratory flow between 25% and 75% of vital capacity [8,9,10,11,13,16,18,19,20,21,24,25].

**Table 1 pathogens-11-01538-t001:** Characteristics of the included studies.

Study Author (Year)	Country	Study Design	Total Number	Male /Female	Onset Age of PIBO, Mean (SD), or Median (IQR)	Age at the Diagnosis of PIBO, Mean (SD), or Median (IQR)	Age at Performance of PFT, Mean Age (SD), or Median (IQR)	Investigated Key Parameters of PFTs	Study Quality
Jung, J (2021) [8]	Korea	cross sectional	47	20/27	NA	NA	5.6 (4.4–7.9) yrs	FVC (z-score), FEV_1_ (z-score), FEV_1_/FVC (z-score), FEF_25–75%_ (z-score), PEF (% pred), % change in FEV_1_ after bronchodilator, post bronchodilator FVC (z-score), post bronchodilator FEV_1_ (z-score)	Good
Colom, A (2015) [9]	Argentina	cross sectional	46	25/21	NA	14 (3.0) mo	9.5 (3.0) yrs	FVC (z-score), FEV_1_ (z-score), FEV_1_/FVC (z-score), FEF_25–75%_ (z-score), TLC (%), RV (%), RV/TLC	Good
Michal Gur (2016) [10]	Israel	case control	20	13/7	NA	NA	15.1 (8.3) yrs	FVC (%, z-score), FEV_1_ (%, z-score), FEF_25-75_ (%, z-score), LCI	Fair
Kim, J (2019) [11]	Korea	cross sectional	23	10/13	NA	NA	7.0 (3.3) yrs	FVC (z-score), FEV_1_ (z-score), FEV_1_/FVC (z-score), FEF_25–75%_ (z-score), change in FEV_1_ after bronchodilator (%)	Good
Mattiello, R (2010) [12]	Brazil, Chile	cross sectional	77	50/27	NA	NA	13.3 (95% CI, 12.4–14.0) yrs	FVC (% predicted), FEV_1_ (% predicted), FEV_1_/FVC (%), FEF_25–75%_ (% predicted), TLC (% predicted), RV (% predicted), RV/TLC	Good
Mattiello, R (2008) [13]	Brazil	cross-sectional	20	14/6	NA	NA	11.4 (2.2) yrs	FVC (L), FEV_1_ (L), FEV_1_/FVC (%), FEF_25–75%_ (L), TLC (L), RV (L), RV/TLC (%)	Good
Mosquera, R (2014) [14]	USA	case series	7	5/2	NA	18.0 (2.0–24.0) mo	70 (median) (range, 48–107) mo	FVC (% predicted), FEV_1_ (% predicted), FEV_1_/FVC (%), FEF_25–75%_ (% predicted), RV (% predicted), RV/TLC	Fair
Cazzato, S (2008) [24]	Italy	case series	10	4/6	2.0 (0.9–3.8) yrs	4.3 (0.9–7.3) yrs	4.9 (3.2–7.3) yrs	FVC (% predicted), FEV_1_ (% predicted), FEV_1_/FVC, FEF_25–75%_ (% predicted), RV (% predicted), RV/TLC, DL_CO_ (% predicted)	Fair
Teixeira, M (2013) [16]	Brazil	RCT	30	23/7	NA	NA	10.9 (2.8) yrs	FVC (% predicted), FEV_1_ (% predicted), FEV_1_/FVC (%), FEF_25–75%_ (% predicted), RV (% predicted), TLC (% predicted), RV/TLC (% predicted)	Good
Yu, X (2021) [22]	China	case series	12	8/4	NA	22.0 (14.0–54.5) mo	73.5 (69–87) mo	FVC (% predicted), FEV_1_ (% predicted), FEV_1_/FVC (% predicted), MMEF_25–75%_ (% predicted)	Fair
Mattiello, R (2016) [18]	Brazil	case series	72	55/17	NA	NA	10 (range, 4–17) yrs	FVC (z-score), FEV_1_ (z-score), FEV_1_/FVC (z-score), FEF_25–75%_ (z-score)	Fair
Frohlich, L (2014) [19]	Brazil	case control	16	12/4	NA	NA	15.3 (3.9) yrs	FVC (L), FEV_1_ (L), FEV_1_/FVC (% predicted), RV (L), TLC (L), DL_CO_ (% predicted)	Fair
Aguerre, V (2010) [21]	Argentina	case series	58	45/13	NA	NA	8 (7–11) yrs	FEV_1_ (% predicted), FVC (% predicted), FEV_1_/FVC (%), FEF_25–75%_ (% predicted), TLC (% predicted), RV (% predicted), RV/TLC	Good
Zhang, L (2000) [20]	Brazil	case series	8	NA	NA	NA	NA	FEV_1_ (% predicted), FVC (% predicted), FEV_1_/FVC (%), FEF_25–75%_ (% predicted), % change in FEV_1_ after bronchodilator	Poor
Kim, C (2001) [25]	Korea, USA	Cross sectional	14	NA	NA	NA	NA	FEV_1_ (% predicted), FVC (% predicted), FEF_25–75%_ (% predicted), PEF, TLC, TGV, RV	Poor
Zheng, H (2022) [23]	China	Cross sectional	20	13/7	NA	NA	NA	FEV1 (% predicted), FVC (% predicted), FEV1/FVC (%), MMEF_25–75%_ (% predicted)	Fair

CI, confidence interval; DL_CO_, diffusion capacity for carbon monoxide; FEF_25–75%_, forced expiratory flow between 25% and 75% of vial capacity; FEV_1_, forced expiratory volume in the first second; FVC, forced vital capacity; L, liter; LCI, lung clearance index; MMEF_25–75%_, maximal mid-expiratory flow velocity 25–75%); mo, months; NA, not applicable; PEF, peak expiratory flow; PFT, pulmonary function test; RCT, randomized controlled trails; RV, residual volume; SD, standard deviation; TGV, thoracic gas volume; TLC, total lung capacity; and yrs, years.

## Data Availability

Not applicable.

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
