# Peer review of "Pulmonary Function in Post-Infectious Bronchiolitis Obliterans in Children: A Systematic Review and Meta-Analysis"

_pathogens, 2022, doi:10.3390/pathogens11121538_

Round 1

Reviewer 1 Report

In this work, Lee and colleagues report an overview of the literature on pulmonary function measurement outcomes in children with a diagnosis of post-infectious bronchiolitis obliterans (PIBO). They performed a meta-analysis using random-effects modeling of included studies, which were mostly cross-sectional studies and case series. They find a large heterogeneity across the 13 included studies. Pooled analysis showed pulmonary dysfunction mostly with an expiratory obstructive pattern (FEF25-75% and FEV1). For example, FEF25-75% was 26.8% (>65% is predicted normal) and FEV1 was 53.3% compared to normal age-appropriate reference of > 80%. This would be in line with the current understanding of the pathobiology of this disease affecting mostly the smaller airways. The manuscript is reasonably well written and the study has scientific rigor. However, I have some major concerns about this work including the novelty and conclusions that can be derived.

MAJOR

1. There have been quite some reviews in the literature on this disease with similar conclusions on the pulmonary dysfunction patterns. It is really doubtful whether this pooling of relatively low level evidence data (see further below) with meta-analysis would add to this existing literature. I am thus concerned that the novelty of this work is very limited.

2. The study is a meta-analysis of 13 mostly cross sectional studies, which are known for selection bias. The data were pooled regardless of time between onset of symptoms, PIBO diagnosis and function tests. Not surprisingly, the studies are very heterogenous. With this broad selection we cannot learn anything about dynamic lung function trajectories, possible influence of medication (it is not noted whether patients in the studies received steroids, macrolides etc etc), possible influence of intercurrent infections, possible effect of type of causative pathogen (not noted what was the etiology) etc etc. I understand and appreciate the effort of the authors to pool the data, but I am really doubtful if we can really learn anything from this collection of low level evidence studies.  

3. The title does not mention a systematic review in terms of a systematic search. I believe the authors wanted to do this, however, they limited their search to only 1 search term. This is highly unusual for a systematic search. I would strongly advise to include a medical librarian to assist in obtaining correct MeSH/search terms. With this I would be concerned that the authors have missed some relevant papers: e.g. Rosewich M in Cytokine 2015; Kim in Chest 2001; Haoqi Zheng in Front Ped 2022; Sisman in Eur Resp J 2017; Ya-Nan Li in BMC Ped 2014 and so on.

4. The description of the main objective and hypothesis (missing now) should be improved. In the Introduction on page 2, they describe “to identify the levels of diverse pulmonary function……”. This is hardly a specific objective. The authors should be much more strict and specific here. Please add a hypothesis, as this will make more clear what can be learnt from this study. Also, the by-sentence “….we performed a meta-analysis on the key parameters…..” in the last sentence of the Introduction is a statement on the tool or method to test your hypothesis/reach your objective. This should be included only in the Methods section. These are some points on basic scientific writing.

5. In the Methods the authors describe: “the primary outcome of the present study was to investigate the levels of key pulmonary…..” (page 2, line 31). This is not a primary outcome. This would be an objective. Please be very clear and write according to the basic scientific reporting rules.

6. There are some definitions lacking in the Methods:

- a priori, which type of studies (RCT, observational retro/pros, cross sectional, case series, case reports) were deemed eligible?

- There is no mention of a priori definition of how the PIBO diagnosis should be made. Which international guidelines were used to define PIBO. How can we tell all included studies really had children with PIBO?

- what was decided a priori on mixed adult and pediatric cohorts? Were these studies excluded or data of the children obtained?

7. English grammar and sentence structure needs extensive revisions throughout the entire manuscript.

8.The are many vague sentences in the Discussion. The author need to be much more clear and specific on how this study will help to gain “…a better understanding of the pathophysiologies of PIBO in children” (page 8, line 162), and how this would lead to a better therapeutic plan on an individual basis as stated in the same paragraph and main conclusion on page 9. I seriously doubt whether clinicians with this review in mind change their perspective to monitoring and treating PIBO in children.

9. The study was not pre-registered at PROSPERO.

Reviewer 2 Report

Manuscript under the title "Pulmonary function in post-infectious bronchiolitis obliterans in children: A meta-analysis" by Eun Lee et. al., Authors performed a meta-analysis of the PIBO in children. This study can help to understand better the pulmonary complications in children. However, authors used only one keyword to search databases which can be improved by adding more keywords. Results of the meta-analysis are just narrated in figures and text in a very textual way. Authors have to revise the keywords and present the results in a better way which can give an easy understanding for a common reader. The discussion part is a simple explanation of the meta-analysis findings. Authors can improve it by comparing the previous findings and making sound statements of their analysis of pulmonary parameters and presenting those parameters in the discussion part. The conclusion also need improvements and authors need to add their opinion on the basis of the study findings. 

Minor comments 

Line 12 check it... and followup and can off a prognosis. What is the clear meaning? 

Line 14 revise this sentence. its a confusing statement. 

Line 16 Which feature were recorded mentioned the important ones 

Line 78 Figure legend must be under the figures 

Line 85 and One in Brazil and Chile each

Revise to make it error-free

Improve the Introduction and discussion part 

Citation are very few  

Round 2

Reviewer 1 Report

This manuscript has improved. In particular, the reader has now a better notion of the context/aim and limitations of the study which enable to nuance the findings. Also the methodshave now been described better.

Author Response

Thank you for your positive response. Thanks again.

Reviewer 2 Report

N/A

Author Response

We thank the reviewer for thoroughly reviewing our paper and the positive response for our manuscript.